# Induced Pluripotent Stem Cells, a Stepping Stone to In Vitro Human Models of Hearing Loss

**DOI:** 10.3390/cells11203331

**Published:** 2022-10-21

**Authors:** María Beatriz Durán-Alonso, Hrvoje Petković

**Affiliations:** 1Unit of Excellence, Institute of Biology and Molecular Genetics (IBGM), University of Valladolid-CSIC, 47003 Valladolid, Spain; 2Biotechnical Faculty, University of Ljubljana, Jamnikarjeva 101, 1000 Ljubljana, Slovenia

**Keywords:** hearing loss, human induced pluripotent stem cell (hiPSC), inner ear

## Abstract

Hearing loss is the most prevalent sensorineural impairment in humans. Yet despite very active research, no effective therapy other than the cochlear implant has reached the clinic. Main reasons for this failure are the multifactorial nature of the disorder, its heterogeneity, and a late onset that hinders the identification of etiological factors. Another problem is the lack of human samples such that practically all the work has been conducted on animals. Although highly valuable data have been obtained from such models, there is the risk that inter-species differences exist that may compromise the relevance of the gathered data. Human-based models are therefore direly needed. The irruption of human induced pluripotent stem cell technologies in the field of hearing research offers the possibility to generate an array of otic cell models of human origin; these may enable the identification of guiding signalling cues during inner ear development and of the mechanisms that lead from genetic alterations to pathology. These models will also be extremely valuable when conducting ototoxicity analyses and when exploring new avenues towards regeneration in the inner ear. This review summarises some of the work that has already been conducted with these cells and contemplates future possibilities.

## 1. Introduction

Hearing loss (HL) affects over 1.5 billion people worldwide, of whom around 430 million require some kind of hearing care (WHO estimates, 2022: https://www.who.int/health-topics/hearing-loss: accessed on 17 March 2022); these figures are expected to rise, and it is estimated that by 2050, there will be at least 700 million people who face disabling HL. Such high incidence and the vastly detrimental impact this disorder has on an individual’s quality of life make it imperative to identify routes towards the establishment of effective therapies. Unfortunately, despite an extraordinary effort being displayed in the field of hearing research, no effective treatment has reached the clinic since the advent of the cochlear implant, rendering HL a largely intractable sensorineural deficit. Highly relevant knowledge has been gained on the molecular interactions that govern inner ear development as well as on some of the pathological mechanisms unleashed by an array of genetic mutations, ageing and exposure to damaging agents such as ototoxins and high levels of noise [1,2,3,4]. Key to these and future studies is the availability of appropriate research models [5,6,7]. The mouse is often the model of choice in pre-clinical testing, especially when attempting to reproduce phenotypes caused by genetic mutations [1,8]; however, translating human mutations into the mouse model does not always result in comparable outcomes to those seen in patients [9,10,11]. Furthermore, there are specific studies where the mouse is not adequate as a model organism; this is the case when testing ototoxicity by aminoglycoside antibiotics, in which the guinea pig is widely preferred [12,13]. Many studies are now being conducted on zebrafish, a model that can also undergo genetic modifications and that offers several other advantages, such as a much faster life cycle than that of the mouse and the generation of very large numbers of test organisms that are transparent at their early developmental stages, greatly facilitating their analysis and making the model amenable to high-throughput screenings [14,15,16]; nonetheless, the results obtained on this model must be validated on some mammalian system [17,18,19]. A critical shortcoming to all these models is that none of them are of human origin and inter-species differences may be expected [7,20]; in this regard, highly relevant data are being obtained from studies on the common marmoset, a non-human primate model of cochlear development and genetic HL [21,22,23,24]. There are strong indications that the common marmoset may constitute a more predictive model than the mouse since the time course for cochlear development in the former is much longer than in the latter and may thus more closely resemble the human process. Additionally, although gene expression patterns in this primate are largely conserved with those in human and rodents, inter-species differences have been identified in the expression of a series of deafness-associated genes whose corresponding mouse mutants have failed to replicate the human pathology [25,26]. Overall, these findings may explain the discrepancies recorded with some murine models and point at the common marmoset as a promising model for hearing impairment in humans; very importantly, these animals may now be genetically modified [27,28].

Another alternative to the absolute scarcity of human inner ear tissue that may complement the work on in vivo animal models is the differentiation of accessible human cell types towards an otic fate [29]. To date, the best results have been obtained with human embryonic stem cells (hESCs) and induced pluripotent stem cells (hiPSCs) [30,31,32]. These latter overcome the ethical concerns associated with the use of hESCs and are quickly becoming the cells of choice; they exhibit a similar plasticity to that of hESCs and may yield patient-specific disease models, as well as autologous cells for transplantation therapies that may be developed in the future.

## 2. hiPSC-Based Cultures to Model Inner Ear Development

The inner ear is a highly complex tissue that contains a large variety of cellular subtypes, arranged in a highly organized manner [33,34,35]. Most of the work that has been carried out to establish in vitro models of inner ear development has focused on the two main sensory cells in the cochlea, the hair cells (HCs) and the spiral ganglion neurons (SGNs). However, protocols are now being developed to generate other otic cell types [36,37]. Additionally, the last years have seen a surge in the number of studies that are conducted on recently developed organoid models, 3D cultures that harbour different types of otic cells and that more closely mimic the in vivo situation.

Initial work on the differentiation of a pluripotent stem cell type towards an otic fate was carried out by Heller’s group [38], who demonstrated that murine ESCs (mESCs) can be made to differentiate into inner ear progenitor cell-like cells that resemble an early otic placode stage. To do this, they treated embryoid body (EB)-derived cultures with a combination of EGF, IGF-1 and basic FGF (bFGF), growth factors that play a role in inner ear development. A population of nestin-expressing cells was obtained that expressed the early otic genes *Pax*2, *Bmp*7, *Jagged*-1 and *Otx*-2. Growth factor withdrawal led to a downregulation in the expression of nestin and the otic vesicle markers *Pax*2 and *Bmp*7; it also resulted in an induction in the expression of genes corresponding to intermediate stages of differentiation of inner ear cell types, such as *Math*1, *Brn*3.1, *p*27^Kip1^ and *Jagged*-1, followed by expression of markers typical of the HC lineage, *Myosin* VIIA, *Espin*, *Parvalbumin* 3, and the *a9 Acetylcholine receptor*; importantly, a high proportion of the cells co-expressed the HC master gene *Math*1 and *Brn*3.1, as well as MATH1 and myosin VIIA (MYO7A), pointing to the emergence of HC-like cells in the cultures in a stepwise manner that resembled the in vivo process. Moreover, these cells continued their differentiation towards a HC-like lineage following their integration into the sensory epithelia of otic vesicles in chicken embryos; an upregulation in the expression of MYO7A and the hair bundle protein espin (ESPN) was recorded. A series of studies has since attempted the differentiation of pluripotent stem cells (ESCs and iPSCs, of murine and human origins) to otic cell lineages; for detailed reviews on these studies, we refer the reader to some earlier publications [20,39,40,41]. Within the scope of this review, we address the work that has been carried out using hiPSCs (Table 1), briefly mentioning some results obtained with ESCs.

Following the observations made by the Heller lab, a series of investigations ensued that have used murine pluripotent stem cells to develop protocols for the generation of otic progenitors (OPs) that can then be differentiated towards other otic cell types, mainly HCs and SGNs. Various groups have opted for a stepwise protocol that consists in promoting the formation of ectodermal cells while inhibiting the emergence of mesodermal and endodermal derivatives; this is most frequently achieved by growing the cultures in a 3D format, either as EBs or as neurospheres, and modifying the culture medium composition [44,47,53]. Induction of early otic markers such as PAX8, PAX2, DLX5, SIX1 and E-Cadherin (ECAD) has been confirmed. Addition of IGF-1 to the medium while treating the stem cell cultures with inhibitors of the mesodermal and endodermal lineages anteriorizes the emerging ectoderm and increases the responsiveness of the cells to otic induction by FGF treatment [53,54]. In addition, similarly to the observations made by Li and colleagues (2003), GF withdrawal often leads to further differentiation of the OP populations into more mature sensory cell phenotypes [38]. Differentiation of OPs towards the HC lineage may be promoted by co-culturing stem cell-derived HC-like cells with support cells such as chick or mouse utricle stromal cells [53,55], when they acquire morphological specializations typical of a HC, such as stereociliary bundles at the apical surface, expression of the HC bundle protein ESPN, and become responsive to mechanical stimulation. Nonetheless, current protocols yield cells that resemble immature HCs [53]. A major obstacle to the optimization of this methodology is that most of the protocols include 3D cell aggregation steps that aid in the differentiation process; this comes at a cost since it translates into greater difficulties to control the culture conditions; it also hinders the identification of specific factors and treatment times that would in turn lead to more efficient and reliable differentiation protocols [30,49]. In order to avoid this problem, Azel Zine’s group designed 2D culture-based protocols that yielded hiPSC-derived otic sensory progenitors within a 13-day period; one method consisted in exposing the cells to an FGF3/FGF10 combination all throughout the differentiation procedure [45]. Expression of the key otic/placodal markers PAX8, PAX2, DLX5 and GATA3 was confirmed in the cultures at day 6, with some cells co-expressing DLX5 and GATA3. Sustained FGF stimulation resulted in increased PAX2 and GATA3 expression levels and decreased expression of markers typical of mesendodermal and other lineages. Further differentiation of the otic/placodal progenitors towards a HC-like phenotype was initially attempted by culturing these cells in the presence of EGF and retinoic acid (RA), as performed by Rivolta’s group when differentiating hESC to HC-like cells [30]; an increase in *ATOH*1 expression was observed. Furthermore, treatment with a Notch inhibitor resulted in increased *ATOH*1 and *MYO*7A expression and co-expression in some cells of MYO7A and POU4F3; nearly half of the cells were MYO7A-positive, as opposed to only 5% in the EGF/RA-treated cultures. Nevertheless, none of HC-like cells presented stereociliary protrusions, and these were therefore very immature cells. An alternative method was described by the same group that consisted in maintaining FGF activation while inhibiting the TGFβ and the WNT pathways during the first 6 days of culture; this combination led to a clear increase in the numbers of PAX2-positive cells in the cultures (approx. 55%), as compared to FGF3/FGF10-only treated cultures. Expression of non-neural ectoderm (NNE)/preplacodal ectoderm (PPE) markers (DLX3/5, GATA2/3, EYA1, SIX1) and the OP markers PAX8, PAX2 and SOX2 was induced. To mimic otic development, WNT3A was added to the OP cultures, and a week later, the cultures expressed embryonic HC markers (ATOH1, MYO6, MYO7A, MYO15A, POU4F3, JAG2) [46].

A very recent publication by Saeki and colleagues (2022) describes a protocol that benefits from a monolayer culture format where transient BMP signalling and TGFβ inhibition efficiently lead to hESC differentiation into PPE, while further differentiation of these cultures to a posterior placode fate is carried out in 3D cultures [56]. Heterogeneity of the starting cell population is thus reduced, while the 3D culture system is meant to counteract the tendency of 2D cultures to promote anterior placode fates and reduce the efficiency of downstream differentiation protocols towards otic placodal lineages. By painstakingly testing whole sets of factor combinations and varying their concentrations and application times, this group has demonstrated the relevance of (a) bFGF and RA signalling in the specification of a posterior placodal fate, (b) BMP inhibition to maintain the expression of otic placodal markers, and (c) WNT activation to further promote otic placode/otocyst induction. However, they have also reported heterogeneous cell cultures as well as failed attempts to maintain JAG+/SOX2+ prosensory cells in culture and to differentiate them to HC-like cells; such differentiation was only attained when these cultures were forced to overexpress ATOH1, POU4F3 and GFI1, three factors that have been shown to induce the differentiation of mESCs into HC-like cells [57]. These data underline the difficulties that accompany the establishment of efficient differentiation protocols; there may be “contaminating” non-target cell populations that provide inductive signals and trophic support to the differentiating otic precursors; although these populations hinder the elucidation of factors that drive the differentiation steps, they cannot be removed from the culture while such factors remain unknown [49,56].

Regarding the work that has been conducted to generate HC-like cells, Chen and colleagues (2018) applied two different methods to differentiate hiPSCs into HC-like cells [43]. One method was based on the protocol developed by Ronaghi and colleagues (2014) [54] to generate HC-like cells from hESCs, whereby anterior ectoderm formation was promoted by treating hiPSC-derived EBs with inhibitors of mesodermal and endodermal cells and IGF-1. Otic induction ensued, following attachment of the EBs and an initial treatment with an BMP inhibitor, followed by an activation of WNT signalling, all in the presence of sustained FGF stimulation. Upregulation of the PPE marker *EYA*1 and the OP marker *DLX*5 preceded an increase in *PAX*2 expression. Differentiation towards HC-like cells was attained by GF withdrawal and changes in the concentration of knockout serum replacement in the culture medium, up to 42 days in vitro; increased expression of the *MYO*7A and the *ESPN* genes and uptake of the FM1–43 dye, frequently used to indicate the presence of mechanotransduction channels in HCs, were reported. Although the researchers observed some ciliary-like protrusions in some cells, these consisted of very few misarranged cilia, resembling nascent HCs. A second protocol used by the group consisted in overexpressing in hiPSCs the transcription factors ATOH1 and two of the regulatory factors for the X-box, RFX1 and RFX3, to drive their differentiation to HC-like cells [43]. RFXs are necessary for correct ciliogenesis and hearing in mice; by using hiPSCs that carried a *MYO*7A reporter construct, these authors were able to monitor the effect of co-overexpressing varying combinations of *ATOH*1 and the *RFX* factors on the differentiation towards a HC lineage. An increase in the numbers of MYO7A-positive cells was identified in *ATOH*1-and *ATOH*1/*RFX*1/*RFX*3-overexpressing cultures; ESPN expression and a higher density of stereociliary-like protrusions that resembled HC bundles were also observed in the latter. Whole transcriptome analysis has shown that overexpression of these three transcription factors leads to an upregulation of genes involved in the formation of HC bundle stereocilia, which could be promoting the progression of hiPSCs towards a HC fate.

A simplified stepwise differentiation protocol had been earlier proposed by Ohnishi and colleagues (2015) [44]. Culture of hiPSCs during 8 days in serum-free medium resulted in over 95% of the cells in the culture co-expressing the PPE markers SIX1 and ECAD; subsequent treatment of these highly homogeneous cultures with the otic inducer bFGF for 15 days led to induced expression of the otic placode marker PAX2 in a very low percentage of the cells, around 0.05%. The growth factor was then removed, and the cultures were grown in the presence of Matrigel to promote differentiation towards a HC-like cell phenotype. Very low induction rates were recorded since only approximately 0.01% of the cells expressed the HC protein MYO7A and an even smaller number of these cells showed stereocilia-like protrusions at their apical side; no kinocilia-like structures were observed in any of the cells. Neuron-like cells were also present in the culture, as indicated by βIII-Tubulin staining.

More recently, Marcelo Rivolta and his group have used lentiviral- and mRNA-mediated reprogramming to obtain hiPSCs and have employed these two types of hiPSC lines to replicate their earlier work on hESCs [42]. Differentiation to OPs was attained by growing hiPSC monolayers in the presence of FGF3 and FGF10, as previously described in [30], and it was accompanied by increased expression of the markers PAX8, PAX2, SOX2, FOXG1; co-expression of PAX8 with any of the other markers was observed in ≤20% of the cells. The progenitors could be maintained in culture under proliferating conditions and presented two different morphologies: flat, epithelial-like islands (otic epithelial progenitors, OEPs) or small cells with cytoplasmic projections (otic neural progenitors, ONPs). Progression of OEPs towards a HC-like lineage was promoted by removing the FGF stimulation and treating the cultures with EGF and RA, a combination that promotes HC differentiation of human foetal auditory stem cells (hFASCs) and hESC-derived OEPs [30,58]. Accordingly, expression of the HC genes *ATOH*1 and *POU*4*F*3 was observed, with co-expression of the two proteins in a proportion of the cells. Electrophysiological recordings pointed at similarities to vestibular sensory cell types. Of interest, no significant differences were observed between the hiPSC lines that had been obtained through mRNA reprogramming and those that had been lentivirally transduced. Other laboratories have followed this differentiation protocol [59,60].

Initial work to differentiate hiPSCs into auditory neuron-like cells was carried out by Gunewardene and co-workers (2014), who applied a protocol used to generate neural crest (NC) progenitors from hESCs, since development of the NC and the otic placode are closely related [61]. Neurospheres were obtained following non-adherent culture in the presence of serum-free medium containing EGF and bFGF; subsequent attachment of these floating cultures on human fibroblast feeder layers and posterior GF withdrawal to promote neural differentiation resulted in cultures that expressed the dorsal hindbrain marker PAX7 and the otic placode marker PAX2, as well as a series of other markers such as SOX2, Islet1 (ISL1), BRN3A and NEUROD1, key players in sensory neuron differentiation. Bipolar neurons that expressed the neural markers βIII-tubulin and NFM were seen emanating from the periphery of the attached neurospheres; also, hiPSC-derived neurons expressed GATA3 and VGLUT1, indicative of a glutamatergic phenotype, and thus resembled type I auditory neurons. Electrophysiological recordings demonstrated that these cells were functionally active. Of note, a higher variability in marker expression was recorded for the various hiPSC lines tested, compared to hESC controls. Decreased potential of the hiPSC-derived neurons to establish contact with denervated HCs in organotypic cultures of cochlear explants was also observed, in comparison to hESC-derived sensory neurons [48]. Nonetheless, hiPSC-derived cells extended processes towards the HCs and appeared to innervate both IHCs and OHCs, resembling early stages of development. Expression of the synaptic marker Synapsin 1 was observed in the regions of contact with the HCs. hiPSC-derived neurons that were at an earlier stage of differentiation established more numerous synapses than the more differentiated neurons [48].

As mentioned above, Boddy et al. (2020) described the differentiation of hiPSCs into two types of OPs, OEPs and ONPs [42]. ONPs give rise to bipolar sensory neurons following sequential treatment with bFGF and then bFGF together with the neurotrophins NTF3 and BDNF; SHH is applied to the cultures during the first 5 days. These neurons express βIII-Tubulin, BRN3A, NEUROD1, NEUROG and are responsive to electrical stimulation. Successful generation of sensory neuron-like cells from hiPSC-derived ONPs using the same protocols that have been applied to differentiate hFASCs and hESCs to these lineages [30,58] is of high relevance, since transplantation of neurons obtained from the differentiation of hESC-derived ONPs into a gerbil model of neuropathic deafness resulted in a certain degree of recovery of the auditory function, starting 4 weeks post-transplantation. Such functional improvement appeared to correlate with the presence of an ectopic spiral ganglion in the animals that contained βIII-Tubulin-expressing cells that extended projections to HCs in the organ of Corti and that appeared to establish synaptophysin-positive contacts in the cochlear nucleus [30]. Other groups have applied the same protocol to generate SGN-like cells from hiPSCs [62].

Following up on their recent publication [56], Kurihara and co-workers (2022) have described a stepwise protocol to produce hiPSC-derived SGN-like cells that integrates 2D and 3D cultures [50]. hiPSCs are initially grown as monolayers in the presence of FGFs, a transient activation of the BMP pathway and a posterior WNT activation, giving rise to homogeneous OP cultures; GFs are then removed, and the cells are shortly grown under hypoxic conditions. It is at this stage that floating cultures are prepared, a new combination of GFs (EGF, bFGF, IGF1, WNT3A, CHIR99021) added to the medium and the cells grown under hypoxic conditions for 5 more days. Thereafter, the cultures are transferred to neurotrophic factor-containing medium under normoxic conditions for at least one month; the resulting cultures have been termed otic organoids and contain SGN-like cells on their surface. Very importantly, these auditory neuron-like cells exhibit morphological characteristics and a protein expression profile that are similar to those of primary SGNs; they also demonstrate electrophysiological activity. Their significance as in vitro auditory neuron models has been demonstrated in ototoxicity assays, as discussed below. Although it is a more laborious method, the generated otic organoids also contain HC-like cells, as indicated by the expression of MYO6 in these cells and the co-expression of ATOH1 and POU4F3 proteins [50]. This is the greatest advantage organoid cultures offer to the fields of inner ear development and hearing research: the possibility to create more complex cellular systems that closely resemble the living tissue, where different lineages co-exist and interact with each other and with the elements that are present in the extracellular matrix.

A major breakthrough in the inner ear research field has been a series of publications by Eri Hashino’s group on the generation of stem cell-derived inner ear organoids [31,63,64,65]. hiPSC-derived inner ear organoids were first described in 2017 [31]; aggregate cultures were differentiated in a stepwise manner and were initially driven to NNE. Differently from hESCs that only required TGFβ inhibition to induce the expression of NNE markers, low concentrations of BMP4 had to be additionally applied to the hiPSC cultures for them to express NNE markers (TFAP2, ECAD). Differentiation towards posterior placode/otic-epibranchial progenitors (TFAP2, PAX8, ECAD, SOX2) was attained by growing the cultures in the presence of bFGF and a BMP inhibitor. Subsequent culture in minimal medium and using Matrigel to provide extracellular support yielded epithelial protrusions that expressed PAX8, PAX2, SOX2 and JAG1 and that resembled otic pits; activation of the WNT pathway further promoted differentiation. Aggregates were kept as self-organizing cultures, and otic vesicles formed from day 18 onwards; supporting cell (SC)- and HC-like cells were observed by days 40–60. The latter expressed MYO7A, PCP4, ANXA4, SOX2, CALB2 and exhibited stereociliary bundles at their apical surface that were positive for the HC bundle protein ESPN; they also had a kinocilium. Electrophysiological measurements demonstrated that the HC-like cells were functional. Together with SC- and HC-like cells, Koehler et al. (2017) also observed the presence of neurons that expressed βIII-Tubulin and BRN3A after 20–30 days of differentiation, and S100β+ cells that resembled myelinating Schwann cells [31]. Of note, there appeared to be synaptic contacts between the neurons and some of the HCs, attending to the expression of various ribbon and synaptic markers. The data obtained by the group indicated that the HC-like cells that were generated within the organoids resembled vestibular HCs. A later publication by Jeong and colleagues (2018) reported the emergence in hiPSC-derived inner ear organoids of functionally active vestibular and cochlear HC types; these researchers had applied a series of modifications to the original protocol such as initial culture of the hiPSCs on a layer of mitotically inactivated mouse embryonic fibroblasts, some changes in the number of cells used to generate aggregates, and some medium modifications [52]. Work is underway to further support the great potential inner ear organoids pose for the field [20,51,66,67]. They have already been used to model inner ear pathology caused by genetic mutations [66].

Although most differentiation work has aimed at producing HC- and SGN-like cells, a large proportion of HL cases are due to alterations in other cellular types within the cochlea, and there is therefore a need to develop appropriate models for these cells. Mutations in the gap junction beta 2 (*GJB*2) gene, which codes for the connexin 26 (CX26) protein, are responsible for a very large proportion of the cases of non-syndromic HL in the world; this protein is found in gap junction channels in various non-sensory cell types in the cochlea, and it plays a role in K^+^ recycling and cochlear homeostasis. Fukunaga and colleagues (2021) differentiated hiPSCs to CX26 gap junction-forming cells that exhibited similar characteristics to SCs [68]. To do this, they prepared SFEBq (serum-free floating culture of EB-like aggregates with quick reaggregation) cultures to initiate differentiation. Differently from the method established by Koehler and co-workers (2017) [31] to generate HC-like cells within human inner ear organoids, Fukunaga et al. (2021) observed that TGFβ inhibition during the induction process led to reduced appearance of CX26-positive cells within day 7 aggregates [68]. Therefore, they only applied BMP and saw an increase in the expression of the otic progenitor markers PAX2, PAX8, GATA3, GJB2 and GJB6. Moreover, this group observed that addition of insulin to the cultures in the presence or absence of BMP was sufficient to promote their growth and survival and the induction of CX26-expressing vesicles within the aggregates; in those cells, CX26 protein was localized to gap junctions at the cell–cell borders. Subsequent culture of CX26-expressing vesicles on a layer of murine cochlear feeder cells yielded proliferative human gap junction-forming CX26-positive cells that resembled cochlear SCs, according to their mRNA and protein expression profiles [68]. Also important is the hiPSC-based model developed by Hosoya et al. (2017) that gives rise to outer sulcus cell (OSC)-like cells [36]. Mutations in the *SLC*26*A*4 gene that codes for the anion exchanger protein pendrin cause Pendred Syndrome (PDS), the most common type of syndromic hereditary HL. Differentiation to OSC-like cells was carried out on monolayer cultures by transiently activating the BMP pathway while growing the cells in a medium that contained a cocktail of FGF factors; OPs were obtained by day 12, when over 80% of the cells in the culture co-expressed PAX8 and FOXG1; expression of other OP markers, PAX2, SOX2, OTX1, GATA3 and TBX1, was also observed. OPs were expanded for 2–4 weeks in a low-serum medium with low bFGF concentrations. Differentiation to a mature OSC phenotype was carried out by changing the cells to a weakly alkaline medium containing 10% serum and NaHCO_3_. Nearly 100% of the cells in the culture expressed high levels of pendrin; expression of mature cochlear OSC markers (e.g., *KIAA*1199, *CRYM*, *DFNA*5, ATP6B1, CX26, CX31, AQP4) was also demonstrated. Chloride-dependent changes in the cells’ pH demonstrated that they were functional [36].

In addition to the work described above, hiPSCs have been differentiated to other cell types that are associated but that are not necessarily restricted to the inner ear, such as NC cells [69] and Schwann cell precursors [70]. Of note, Wakizono et al. (2021) have reported that Schwann cell precursors proliferate following ouabain-induced damage to the SGNs of adult mice; they have also shown that administration of EGF and bFGF leads to increased proliferation of these precursors [71]. Co-injection of these GFs with the histone deacetylase inhibitor valproic acid results in a significant increase in PROX1+ type 1 SGNs that appear to originate from the proliferating Schwann cell pool and in a partial recovery of the hearing function in ouabain-treated animals. Damage hiPSC-based models that contained both the SGN and the Schwann cell types could shed light on the potential of human Schwann cell precursors to transdifferentiate into SGNs and on the signals that modulate such processes. The availability of protocols to differentiate hiPSCs to NC cells aaaais also of interest in inner ear research, since it is known that cranial NC cells integrate during development into the otocyst and give rise to melanocytes of the stria vascularis; a possible NC origin of some glial cells in the cochleovestibular ganglion [72] and even of some SGNs has been postulated.

## 3. hiPSCs to Generate Genetic Models of HL

A genetic component has been identified in a large number of HL cases, not only in the form of hereditary congenital HL but also in relation to age- and noise-induced hearing impairment [73,74]; more than 150 genes have already been associated with non-syndromic HL and with over 400 forms of syndromic HL; this number of genes is expected to rise [1,75,76,77] (http://hereditaryhearingloss.org: accessed on 17 May 2022). Hurdles to the identification of some of these genes are the high genetic heterogeneity of HL [1,2] and the diversity of cell types that may be affected by different mutations [76,77,78], the existence of mutations that are not always identifiable by commonly used techniques such as whole exon sequencing (WES) [75,79,80,81,82], and the varying length of time to the onset of symptoms, among others [62,83]. The identification of HL-related genes is highly dependent on the existence of appropriate models where a causative role for the mutation can be demonstrated, and correction of the mutation may be directly correlated to some restoration of the physiological functions. In this regard, the amenability of the mouse to genetic modifications makes it the model of excellence in this type of study, especially when coupled to current gene editing systems such as CRISPR/Cas9 and base editors [8,84,85]. In addition to the validation of suspect genes, the mouse has also been used to identify genes whose association to HL was not predicted; an example of this is the work carried out by Ingham and colleagues, who identified 38 novel deafness genes following a large-scale screen of 1211 mouse mutant lines [2]. Of note, 11 of the genes identified within the scope of this work have now been associated with hearing capacity in humans. Nonetheless, there are important differences between humans and mice that affect the potential of these latter to serve as models for HL in man [86]; they present a shorter lifespan that may preclude them from appropriately modelling the late onset and progressive HL recorded in many patients, as well as a lack of genetic heterogeneity that is in stark contrast to that found in human populations [1,87]; they also differ in the time that is required for their inner ears to fully develop and in the amino acid sequence of important proteins [88]. Moreover, there are mouse lines that, despite carrying genetic mutations identical to some found in patients, do not exhibit the same symptoms [9,10,87]; in this regard, differences in gene expression patterns between primates and rodents have been associated with a failure of some of the murine models to recapitulate the human disease [9,21,24,25,26]. The common marmoset, also amenable to genetic modification, has thus been proposed as a more predictive model of human cochlear development and pathology [21,22]. In addition, hiPSC lines are being generated from somatic cells obtained from patients with HL [89,90,91,92,93] and subsequently differentiated towards a range of different otic cell types [37,59], thus providing human cell-based models that overcome the limitations imposed by the complex anatomy of the inner ear and the lack of human tissue; very importantly, these cultures retain the genetic backgrounds and the mutations present in patients with hearing deficits, constituting an ideal substrate for studies on the molecular mechanisms that lead to syndromic or non-syndromic HL and the validation of therapeutic targets through the creation of isogenic control cell lines [59,88,94]. Additionally, they provide the means to model mutations within large non-coding chromosomic regions that would otherwise be very difficult to replicate. An example are the deletions identified by Bademci et al. (2020) in a region downstream from the **Growth and Differentiation Factor 6 *(GDF*6)** coding sequence [82], associated with non-syndromic cochlear aplasia; hiPSCs and hiPSC-derived cultures of otic lineage precursors were established from deaf individuals, and a range of molecular analyses indicated that the mutation correlated to decreased GDF6 expression levels; moreover, cochlear aplasia has been subsequently observed in *Gdf*6 KO mice [82].

Using hiPSCs, a key role was demonstrated for **MYO7A** [59] and **MYO15A** [60] in the formation of stereocilia in HCs. Tang and co-workers compared HC-like cells obtained from hiPSCs derived from a hearing-impaired patient carrying mutated *MYO*7A alleles to those differentiated from the same hiPSCs following the correction of one of the mutated sequences; gene editing resulted in morphological recovery of the stereociliary arrangement and the restoration of mechanotransduction channel activity; this group thus reported a requirement for the MYO7A protein in the assembly of stereocilia into HC bundles [59]. Similarly, Chen and colleagues (2016) proved that the MYO15A protein was necessary for the normal elongation and organization of the stereocilia in HC bundles during HC differentiation, since HC-like cells differentiated from hiPSCs derived from a patient carrying mutations in this gene exhibited F-actin disorganization and abnormally shortened stereocilia, coupled to altered electrophysiological recordings [60]; these alterations disappeared following correction of one of the *MYO*15A sequences [60]. hiPSC lines and corrected isogenic controls have also been generated from patients with mutations in other genes that affect HC development, such as the **Transmembrane Channel-like protein isoform-1 (*TMC*1) gene** that encodes for a key protein of the mechanotransduction channel [90,95]; Wang et al. (2020) have generated an hiPSC line carrying a *TMC*1 c.1253 T > A mutation that constitutes the human counterpart to the Beethoven mouse model [96]; gene editing of the *Tmc*1 mutation has been shown to significantly ameliorate the progressive HL seen in these mice [34,97,98]. Another hiPSC line has been created by Dykxhoorn and co-workers from a patient with a mutation in the **Small Muscle Protein X-linked (*SMPX*) gene** that may serve as a model of X-linked deafness 4 (DFNX4) [92]; increased susceptibility to loud noise and progressive HL have been observed in *Smpx* KO mice, associated with progressive degeneration of stereocilia in the OHC bundles [99]. hiPSC lines have also been established from patients with syndromic HL such as those carrying mutations in the ***USH*2A gene** [94,100,101,102], accountable for a large number of Usher Syndrome cases, a highly heterogeneous disorder that is mainly characterised by vision and hearing impairments [83,103]. The *USH*2A gene encodes the protein Usherin, a constituent of the ankle links that connect adjacent stereocilia during the formation and maturation of HC bundles [103], and alterations in its expression lead to moderate-to-severe HL. While correcting one or both copies of a c.2299delG mutation in hiPSCs derived from an Usher Syndrome type 2 patient, Sanjurjo-Soriano and colleagues (2020) observed that this mutation leads to increased levels of mutant *USH*2A mRNA, therefore escaping the nonsense-mediated mRNA decay pathway that rids the cell from the toxic accumulation of truncated proteins; CRISPR/Cas9-mediated editing of the mutant *USH*2A sequence resulted in expression levels that were similar to those in control hiPSCs [102]. Very relevant was also their observation that a milder pathogenic variant of the *USH*2A gene, c.2276G > T, did not correlate with alterations in the levels of mRNA expression; moreover, compound heterozygous hiPSCs that carried a c.2276G > T change in one allele and a c.2299delG mutation in the other exhibited normal mRNA expression levels, lower than those registered in hiPSCs carrying a c.2299delG mutation either in one or both *USH*2A alleles [102]. Waardenburg Syndrome (WS) is the most common form of syndromic HL, characterized by pigmentation anomalies and hearing deficits; contrary to Usher Syndrome, it is inherited in an autosomal dominant form. Wen and co-workers (2021) have recently reported the establishment of hiPSC lines from WS patients who carry mutations in genes that have been associated with the disease; thus, hiPSC lines are now available that carry mutations within the **Paired Box Gene 3 *(PAX*3) gene** [104,105] and the **SRY-Box Transcription Factor 10 *(SOX*10) gene** [69] that play key roles during the development and differentiation of NC cells and the formation of the stria vascularis in the cochlea. Differentiation of hiPSCs containing mutant *SOX*10 sequences has allowed for the identification of alterations in the expression of genes that play a key role in NCC specification and the confirmation of a deficient differentiation of these cultures to NCCs [69].

hiPSCs have also been created that carry mutations known to affect otic cell lineages other than HCs, such as the auditory neurons and a variety of SC types. Otic sensory neuron-like cells differentiated from hiPSCs from patients with an autosomal dominant mutation in the **Microtubule-Associated Protein 1B *(MAP*1B) gene** exhibit altered microtubule dynamics, markedly reduced neurite lengths and electrophysiological impairments [62]; this phenotype is due to reduced expression and deficient phosphorylation of the MAP1B protein, involved in the regulation of microtubule stability and dynamics and the function of otic sensory neurons. These abnormalities have been observed in SGNs of *Map*1b-KO mice that show late-onset progressive SNHL [62]. Li et al. (2021) have also reported the creation of an hiPSC line from a patient carrying a mutation in the X-linked **Apoptosis-Inducing Factor Mitochondria associated 1 *(AIFM*1) gene** [106]; a large number of mutations have been identified in the *AIFM*1 gene that are associated with late-onset auditory neuropathies, characterized by impaired speech perception [107]. AIFM1 participates in the maintenance of mitochondrial homeostasis and in caspase-independent death processes. The AIFM1 protein is widely expressed in the murine inner ear, mostly in the cytoplasm of IHCs, OHCs and SGNs; however, mutations in the *AIFM*1 gene have been linked to late-onset degeneration that does not affect OHC function [107].

Several hiPSC lines are also available to study the detrimental effects of certain mutations on the physiology of otic SC types. A number of these carry mutations in the **Gap Junction Beta 2 *(GJB*2) gene** that codes for the CX26 protein and is associated with over half of the cases of hereditary HL in the world with more than 340 pathogenic variants [68,89,91,108,109]. Fukunaga and colleagues (2021) have recently reported that absence of the CX26 protein in gap junction-forming inner ear SC-like cells generated from patients’ hiPSCs results in deficient gap junction-mediated intercellular communication, associated with shortened gap junction plaques in those cells [37]; in this case, the authors have reported similar findings to those obtained in *GJB*2 mutant mice [110]. hiPSC lines have been created from patients who carry mutations in another gene that is expressed in cochlear SCs, the **Transmembrane Protein 43 *(TMEM*43) gene** [111]. It codes for a protein that physically interacts with CX26 and CX30 and participates in the regulation of passive conductance current and K^+^ recycling in the cochlea; mutations in the *TMEM*43 gene lead to a late-onset auditory neuropathy spectrum disorder and progressive HL in humans. Progressive HL has also been reported in mice harbouring the mutated human *TMEM*43 sequence [112]. On the other hand, other types of HL, such as PDS, lack appropriate animal models [10,113], and patient-derived hiPSC lines will therefore provide highly valuable information [36,114,115]. Although no progressive deafness is observed in any of the rodent models for PDS [9], a degenerative phenotype has been described by Hosoya and colleagues on outer sulcus cell (OSC)-like cells derived from hiPSCs generated from PDS patients with mutated **Pendrin/*SLC*26A4 (Solute Carrier Family 26 Member 4)** alleles [36]. While studies on mice carrying similar mutations pointed to a loss of function of the anion exchanger pendrin as the cause for this disorder, correction of the mutation in the human model has demonstrated that this is instead the result of the intracellular accumulation of aggregates of the mutated protein that lead to increased susceptibility to cellular stress; these findings have pointed to the activation of autophagy as a possible therapeutic avenue [36]. Moreover, the present model can better explain the progressive and sporadic nature as well as the variability of the symptoms observed in PDS patients, and it strongly supports the need for human models of disease.

Patient-derived hiPSCs may be used to explore the effect that mutations in some genes may exert not only on one but on various otic cell types. This is the case with hiPSC lines that carry mutations in the **Epithelial Splicing Regulatory Protein 1 (*ESRP*1) gene** that codes for an RNA-binding protein involved in the regulation of alternative splicing in epithelial tissues. Aberrant splicing of a series of transcripts has been shown in hiPSCs generated from patients with SNHL who carry pathogenic mutations in both *ESRP*1 copies; the alterations disappear following correction of one of the alleles [79]. This phenomenon has also been observed in *Esrp*1 KO mouse embryos, affecting inner ear morphogenesis, and causing delayed auditory HC differentiation and erroneous cell fate specification within the cochlear lateral wall. Therefore, these hiPSCs constitute promising models to study processes involved in the specification and differentiation of sensory and non-sensory cell types in the cochlea and the elucidation of processes leading to HL. In addition, Dong and colleagues (2019) have established patient-derived hiPSCs that carry a mutation in the **Purinergic Receptor p2x *(P2RX*2) gene**; this gene codes for the P2X2 protein, which forms ATP-gated ion channels [88]. Mutations in *P2RX2*, inherited in an autosomal dominant fashion, have been associated with progressive SNHL and increased susceptibility to noise- and age-related hearing loss [116]; however, the pathophysiological mechanism has not yet been elucidated. Since P2X2 participates in a wide range of cellular processes and is expressed in multiple cochlear cell types (HCs, SCs, SGNs), differentiation of these hiPSCs to different otic cell lineages will help to unveil the mechanisms through which aberrant *P2RX*2 expression leads to HL. Other mutations have been described in mitochondrial genes that have been associated with an increased susceptibility to presbycusis progression, noise- and ototoxin-induced SNHL [116,117,118]; most frequently, the development of symptoms depends on the convergence of these mutations with other factors such as exposure to ototoxins and/or high levels of noise, or the presence of other modifier genes [119,120]. Hsu and co-workers have established an hiPSC line that carries an A1555A > G mutation in the **Mitochondrial 12S Ribosome RNA *(MT-RNR*1) gene** [93]. This is the most prevalent mutation in *MT-RNR*1, and it results in increased binding of aminoglycoside antibiotics (AGs) to the mitochondrial ribosome and AG-induced HL; it is therefore important to identify MT-RNR1 variants that may render individuals more likely to develop AG-induced HL prior to the administration of this type of anti-infective [121]. In turn, hiPSC lines such as the one developed by Hsu and co-workers (2017) could become very useful tools in screening programs to evaluate the ototoxic potential of newly developed AGs [93]. Highly illustrative of the complexity of the interactions leading to hearing impairment is the work conducted by Chen and Guan (2022); these authors generated hiPSCs from individuals who either carried an A1555A > G mutation or that same mutation coupled to a mutation (c.28G > T) in the **TRNA5-Methyaminomethyl-2-Thiouridylate methyltransferase *(TRMU)* gene** [119]. The TRMU protein is a tRNA modifying enzyme, and alterations in its expression result in increased sensitivity to AG-induced damage and concomitant mitochondrial dysfunction when these drugs are administered [122]. Chen and Guan (2022) have demonstrated that a mutation in one of the *TRMU* alleles aggravates the severity of the *MT-RNR*1 A1555A > G mutant phenotype, since the differentiation of such hiPSCs to otic epithelial progenitors and to HC-like cells is hindered compared to that of hiPSCs that have undergone CRISPR/Cas9-mediated correction of the *TRMU* mutation [119].

A list of hiPSC lines that have been derived from hearing-impaired patients is shown in Table 2. They most often correspond to mutations that have been identified in a small number of individuals, highlighting the vast heterogeneity that accompanies the genetics of HL [34]. Nonetheless, these lines are all highly valuable since not only do they allow for the validation of specific mutations as the cause of disease, but they may also shed light on more general mechanisms leading to HL that may then become new therapeutic targets [36]. Moreover, tailor-made hiPSC-based models may also be generated using current gene editing tools such as the CRISPR/Cas9 system to introduce specific mutations, as already shown in mESCs [66].

## 4. hiPSC-Based Drug Screening Systems

A plethora of factors may target the various OPs and the differentiated cell types in the cochlea, leading to hearing impairment; among these are ageing, high levels of noise, exposure to ototoxins, infections and, as discussed above, genetic factors. Highly valuable information has been obtained from studies on animal models regarding these processes. However, there are important considerations when selecting an appropriate model organism to evaluate the ototoxicity of a drug lead or the otoprotective potential of a compound [12,13]. The mouse, highly amenable to genetic modification, is frequently the model of choice [2]; ototoxicity studies are often carried out on immortalized murine otic cell lines that allow for high-throughput screenings [130,131], and on cochlear explants, that are considered to be closer mimics to the in vivo situation [3,132]. Nonetheless, differences have been observed between the various tests, as well as among animal models [13]; for example, Ishikawa and colleagues (2019) reported important discrepancies when testing the ototoxic potential of a series of aminoglycoside antibiotics on murine otic cell lines, cochlear explants, and an in vivo guinea pig model, much preferred to the mouse when testing this type of antibiotic [12,13] (Murillo-Cuesta et al., 2010; Poirrier et al., 2010). In addition, work carried out by Oishi and co-workers (2014) revealed a protective effect of metformin against gentamicin-induced toxicity on murine cochlear explants, while co-administration of these compounds in guinea pigs did not prevent HL [133]; Shoman et al. (2018) reported a different response of mice and guinea pigs to ouabain, with SGN ablation and OHC preservation in the former and significant damage to both cell types in the latter [134]. At present, extensive work is being conducted on zebrafish. Damage to the HCs located in the lateral line of zebrafish larvae is easily assessed and is used as an indicator of ototoxicity; this fact, together with the large numbers of animals that can be generated, make the zebrafish an excellent model for high-throughput analyses, frequently aimed at the identification of possible otoprotectants [135]; nonetheless, there are important differences between the mammalian inner ear and the lateral line in zebrafish, and therefore, the results that are obtained on this latter must be subsequently validated on mammalian models [17,136]. On the other hand, the establishment of protocols to differentiate hiPSCs into a range of otic cell types [31,36,50,114,137] have made these cultures highly valuable models to study processes that lead to human inner ear pathology and that may differ from the observations made on animal models; it has also paved the way for the development of hiPSC-based screening platforms that enable the identification of otoprotective drug leads. Establishment of these screening systems conveys the availability of highly homogeneous cultures where most cells have differentiated towards the desired cell type. A good example is the protocol developed by Hosoya and colleagues (2017) to obtain highly homogeneous cultures of cochlear OSCs from PDS patient-derived hiPSCs; such a system has allowed for the validation of the autophagy inducers rapamycin and metformin as otoprotective agents, setting up the stage for further drug screening efforts [36]. In a follow-up study using hiPSC-derived OSC cultures, the same group showed that concentrations of rapamycin much lower than those used clinically were sufficient to attenuate the vulnerability of the cells to both acute and chronic stress conditions through the activation of autophagy; unexpectedly, improved cell survival rates did not correlate to a reduction in the formation of pendrin aggregates, and therefore, further studies will be required to accurately pinpoint disease pathogenesis [129]. Importantly, another mTOR inhibitor, sirolimus, has been identified that exerts a protective effect on PDS hiPSC-derived OSCs at a much lower dose than what is already administered in the clinical setting; Keio University in Japan has been conducting a phase I/IIa clinical trial to test the safety and the effects on the audiovestibular function of administering sirolimus tablets to a group of sixteen PDS patients (Trial registration number: JMA-IIA00361) [9].

Besides using hiPSC cultures that are efficiently differentiated to a specific cell type, more heterogeneous cultures may be employed in drug screening work by specifically labelling the cell type of interest. An example is the work carried out by Kurihara and colleagues (2022), who have recently established 3D hiPSC-derived otic cultures that contain neuron-like cells on their surface that closely resemble primary SGNs in their morphological and electrophysiological properties [50]. These researchers have developed a method to assess the toxicity on the SGN-like cells of the Na^+^/K^+^-ATPase inhibitor ouabain that selectively ablates SGNs in rodent models, and of cisplatin, a broadly used chemotherapeutic agent; auditory neuron-like cells are fluorescently labelled by transducing them with an adeno-associated virus that carries the EGFP sequence under the control of the human synapsin 1 promoter. Toxicity of the compounds is evaluated as a reduction in the intensity of GFP fluorescence that is recorded during whole-mount live-cell imaging; these cultures constitute the first hiPSC-based model to test the effects of drugs on SGN-like cells. These researchers have observed that, while cisplatin treatment elicits a range of processes typical of apoptosis, similarly to what has been observed in animal models, an increase in cell area occurs following application of ouabain, with no upregulation of cleaved caspase-3; these latter observations raise the question of whether the responses of human SGNs to ototoxic drugs are identical to those described in animal models [138]. On the other hand, an otoprotective effect has been shown for the cyclin-dependent kinase-2 (CDK2) inhibitor kenpaullone against cisplatin treatment, in agreement with data obtained from animal work [136]; however, this protection was transient and did not ultimately prevent SGN-like cell death.

Important shortcomings such as drug accessibility, line-to-line variability and the high demands that producing high-quality hiPSC lines place on researchers’ time and workload must be overcome to guarantee efficient high-throughput drug screenings. Moreover, protocols must be developed to reliably evaluate the effect of test compounds on hiPSC-based drug-screening platforms. Important advancements are being reported in cell culture processes that are based on the use of robotics to achieve an automated reprogramming, maintenance and differentiation of hiPSCs, and that are contributing to the creation of large and reliable repositories of hiPSC lines [139,140]; equally important are the improvements in image analysis techniques and artificial intelligence technologies that enable the processing of large amounts of datasets, cell identification and the elucidation of a cell’s pathological state based on its morphology [141,142,143,144]. New computational prediction models are being generated, such as those described by Zhang et al. (2020) [145] and Huang et al. (2021) [146] to predict drug-induced ototoxicity, thus contributing to drug design optimization. In addition, new developments in the field of microfluidics will allow for the optimisation of hiPSC-based work in terms of number of cultures and tests that can be reliably processed at a given time while minimizing the amounts of reagents required, of special relevance when searching for promising drug leads [139,147]. Although such automated processes are not currently accessible to all laboratories, given their technical demands and expense, a wider application is to be expected over time, aided by devices such as that described by [148]. This group has designed two microfluidic devices to test multifactorial cell medium combinations that can be generated from 3D-printed molds and that are easily operated with a single syringe pump; using these devices, they have successfully differentiated hESCs to auditory neuron-like cells. These researchers have reported a much greater homogeneity of the cultures obtained using microfluidics than that observed when applying the same protocol to cultures grown under conventional culture conditions [149]. Application of all these technologies should facilitate the development of large-scale drug screenings [150]. Moreover, the inclusion of 3D hiPSC-based culture models in these searches would be highly desirable, despite posing additional difficulties to the collection and processing of informative data compared to 2D monolayers [151]. Three-dimensional cultures more closely mimic in vivo conditions, in terms of cellular complexity and maturation stage, cell rearrangement within a 3D tissue-like architecture and the establishment of intercellular and cell-matrix interactions, all leading to a cellular context that is closer to the physiological environment [20,31,140]. Not only would a combination of 3D hiPSC-based cultures, the labelling of specific cell populations, automated screening platforms and microfluidics allow for high-throughput screening of candidate compounds but it would also enable more extensive ADME studies (absorption, distribution, metabolism and excretion) on large numbers of derivatives from promising drug leads. Such systems should ultimately reduce the numbers of animals used in experimentation and help overcome the frequently poor predictability of animal models [7,152,153]; this, coupled with greater efforts on compound optimization, should in turn reduce the high attrition rates recorded among candidate drugs advancing to clinical trials [153,154].

## 5. Conclusions and Future Perspectives

hiPSCs have firmly entered the field of biomedical research, as a very attractive alternative to the animal models that are currently used to mimic human development and disease. There is a risk that existing inter-species differences may not be identified, coupled to current trends to reduce the numbers of animals used in research, as stated by the 3Rs principle. hiPSCs are also preferred to hESCs that, although human in origin, are more difficult to obtain and pose important ethical concerns. However, hiPSCs are not without shortcomings. Reprogramming of somatic cells is not always completely efficient, depending on factors such as the cell type and the age of the donor, and this may lead to permanent alterations in the genome [155]; also important is the phenomenon of epigenetic memory [156] that has been observed in these cells and that may hamper their differentiation capacity and the phenotypes of the cells they give rise to. Another concern regarding their use is the way reprogramming is often conducted, that is, through viral transduction of a somatic cell [157]; possible integration of viral sequences in the human cell’s genome conveys a serious risk to the therapeutic use of these cells and remains an important consideration when employing them in research. Alternative reprogramming methods are to be encouraged that overcome this problem, such as the use of synthetic mRNAs, as described by Boddy et al. (2020) [42]. Additionally, variability has been observed in the results obtained with hiPSC lines that has not been recorded with hESCs [47,49]. Efforts are being made to build large hiPSC repositories from healthy and diseased donors; these hiPSC lines are to be generated under a common set of protocols, making use of automated platforms, so that the quality and reliability of the available lines are improved.

Regarding the use of hiPSCs to reproduce pathological states, there is also the question of how to model diseases that, similarly to most cases of HL, are ageing-related. Reprogramming conveys a return to an embryonic status and a permanent loss of age-related markers; ageing traits are not re-established following hiPSC differentiation to desired cell lineages. Various strategies have been contemplated to induce ageing in hiPSC-derived cells [158], such as the overexpression of progerin, the truncated form of lamin A that is associated with premature ageing. Miller et al. (2013) have shown that forced expression of this protein in hiPSC-derived cells restores their original age-related features [159]. Importantly, progerin is expressed at low levels in healthy subjects, and its ectopic expression induces a normal ageing process.

A vast amount of work is being carried out to unravel the complex signalling cues that regulate the differentiation of hiPSCs to otic cell lineages. A very large part of these studies is being guided by data obtained on animal models such as the mouse, the zebrafish and the common marmoset; this latter is shedding light on discrepancies encountered between human and the rodent models. Additionally, 3D cultures are being generated that provide a more permissive microenvironment for differentiation but add variability to the process and hinder the identification of key guiding cues; on the other hand, although the conditions for 2D monolayers are more easily controlled, these cultures show a tendency to generate anterior fates. The development of single-cell analysis techniques and high-throughput transcriptomics are facilitating the creation of detailed maps to differentiation that result from a systematic analysis of multiple markers at many intermediate time points [49,160]. The emerging differentiation landscape is becoming increasingly complex, as temporal and cell-dependent differences are shown to modify the function of single proteins [161] and new cellular subtypes are identified [34]; improved protocols will rest on the identification of yet unknown key guidance cues as well as on a better knowledge of the time windows required for the intervening molecules to be effective and promote differentiation to the desired cell types [162]. We have also learned that aiming for highly pure populations may in fact hamper the differentiation process, since non-target cells may provide the necessary conditions for the culture to differentiate towards the lineage of interest. A key role for the composition of the extracellular matrix in differentiation has also been demonstrated [67,160]. All the gathered knowledge will undoubtedly lead to better protocols for the differentiation of hiPSCs towards otic cell lineages. Furthermore, a convergence of improved cell models such as the inner ear organoids, with current technological developments such as high-content imaging, microfluidics and machine learning, and with powerful gene editing tools such as the CRISPR/Cas9 system, will undoubtedly lead to important advancements in the field of hearing research.

Of great relevance is also the development of human ototoxicity models that help identify possible otoprotectants for the human inner ear. Huge efforts are already being directed at establishing hiPSC-based platforms that support drug discovery work. Cardiotoxicity, hepatotoxicity and neurotoxicity constitute the main reason behind the extremely high drug attrition rates resulting from clinical trials. These are considered to be partly due to a lack of human models; an enormous effort is thus being placed on generating hiPSC-derived cardiomyocytes, hepatocytes and neural cells that may unveil a possible toxicity of the drug leads at early stages of development; parallel to this work is the development of high-content imaging and artificial intelligence systems such as deep learning [163]. Given the very high incidence of HL in humans—often the result of exposure to damaging agents such as some anti-cancer drugs—the identification of otoprotectants is also of the utmost importance. Ototoxicity work is being carried out on immortalized cell lines, murine cochlear explants and live animal models such as the guinea pig and the zebrafish; however, discrepancies have often been recorded between the various models [3,133], and the data obtained on these models has not yet resulted in any benefit for the patients.

There are two other important aspects to the use of hiPSCs in inner ear research that have not been mentioned in this review, i.e., the **transplantation of hiPSC-derived otic cells into the inner ear** and the generation of **hiPSC-based models to explore avenues to cell regeneration within the human inner ear**. Although such applications are of the utmost importance and impressive work has already been conducted, these are newer lines of research that will no doubt constitute a major topic in many publications to come. Intensive work is being carried out to establish methods to introduce exogenous cells into the inner ear in a safe and efficient manner. In addition to the concerns regarding the genetic stability of hiPSCs and the risk of tumorigenicity [164], there are clear obstacles to the transplantation of hiPSC-derived otic cells in the inner ear, namely the survival of exogenous cells in the K^+^-rich endolymph, and the morphological and functional integration of the transplanted cells into the tissue; these issues are being addressed [165,166,167]. Promising data have come from work with ESC-derived neural progenitors [30,168], showing an improved auditory function in transplanted animal models; of great interest are also the results from Lopez-Juarez and collaborators (2019) that demonstrate the migration and engraftment of hiPSC-derived OPs in an ototoxicity guinea pig model, together with an initial differentiation of OPs engrafted in the organ of Corti area towards HC and SC phenotypes [169].

In addition, availability of hiPSC-based models that closely replicate inner ear development in humans offers the opportunity to identify cells within these cultures that are counterparts to cell types that have shown regenerative capacity in mouse models. Such types of cells have already been identified in human foetal tissue [170,171] and in hiPSC-derived OP cultures [46], and a publication by Massucci-Bissoli et al. (2017) has reported the presence of ABCG2-expressing cells in the adult human cochlea [172]. hiPSC-based cultures such as the inner ear organoids constitute an ideal model to study these cells and try to validate the results already obtained on animal models. Various putative progenitor cells have been described in the murine neonatal cochlea; one of these is a population of *Lgr*5-expressing SCs. Albert Edge’s laboratory has gathered a large amount of information on the properties of these cells, which demonstrate regenerative potential in the neonatal cochlea, since they give rise to new HCs following damage to the cochlea. Edge’s group has generated non-adherent cultures of murine LGR5+ cells that can be propagated in culture and used in a variety of in vitro tests, where they have obtained highly promising results [173]. Should human LGR5+ cells possess similar properties to their murine counterparts, the hiPSC-derived inner ear organoids would provide a unique opportunity to study the properties of the LGR5+ population while maintaining these cells within a microenvironment that is much closer to the cochlear tissue than that in the 3D cultures of murine LGR5+ cells. This is highly relevant, in view of the important role that other cell types and the extracellular matrix components have on the behaviour of cells, including the differentiation of LGR5+ cells [174]. Regarding the identification of LGR5+ cells within the organoid, gene editing of the starting hiPSC population could be conducted, as it has already been performed in other systems [175], so that LGR5+ cells and their progeny could be identified and traced. This would enable studies on the effect of different compounds on the proliferative and differentiation potential of these cells, both as isolated cells and within the organoid. Work described by Liu and colleagues (2021) has demonstrated that 3D cultures containing fluorescently labelled target cells may be amenable to high-throughput screening [176]; nonetheless, their 3D culture system is not comparable to the inner ear organoids, of a much greater cellular complexity [176]. Single-cell analyses and high-resolution imaging should facilitate regeneration studies in the latter. There are other shortcomings to the approach described above, such as the vestibular phenotype of the HCs that emerge within the currently established inner ear organoid models; however, the available data support similar responses of the vestibular and cochlear LGR5+ populations. There is also the eternal question of the relevance of the results obtained on an immature system towards their application on the adult organ. As already mentioned, there exists the possibility of inducing “ageing” of in vitro cell-based models; various methods have already been described [159,177].

## Figures and Tables

**Table 1 cells-11-03331-t001:** In vitro inner ear cell models that have been derived from hiPSCs.

HC-like Cells		
Differentiation Method	Main Outcomes	Reference
LV-and mRNA-reprogrammed hiPSC lines Monolayer cultures EGF and RA treatment of hiPSC-derived OEPs	OEPs give rise to HC-like cells (ATOH1, POU4F3) Electrophysiological recordings point to similarities to vestibular sensory cell types Comparable results for LV-and mRNA-reprogrammed hiPSCs	[42]
Two methods: As in Ronaghi et al. (2014), (EBs + attachment + factors) ATOH1/RFX1/RFX3 overexpression in OPs	Improved differentiation to a HC-like cell phenotype in cultures transduced with ATOH1/RFX1/RFX3 viruses. Overexpression of these 3 genes results in HC-like cells with more differentiated morphology (HC bundle-like protrusions) and increased expression of genes involved in stereociliary bundle formation	[43]
Prolonged bFGF treatment to differentiate PPE cells towards an otic placode GF withdrawal to induce differentiation towards HC-like cells	Simple induction method. Culture in serum-free medium very efficiently induces spontaneous differentiation to PPE. Nearly all cells express the PPE markers SIX1 and ECAD Very low otic placode induction rates Very low HC induction rates. Stereocilia-like protrusions in some of the cells. No kinocilia-like structures observed	[44]
Differentiation of Otic/placodal progenitors: FGF + WNT EGF/RA vs. Notch inhibition	EGF/RA-treatment promotes ATOH1 expression. A very low proportion of cells express MYO7A protein Notch inhibition is a stronger promoter of differentiation towards a HC-like phenotype (ATOH1, MYO7A, POU4F3 expression) than EGF/RA-treatment The generated HC-like cells lack stereociliary formations	[45,46]
**SGN-like Cells**		
**Differentiation Method**	**Main Outcomes**	**Reference**
LV-and mRNA-reprogrammed hiPSC lines Monolayer cultures bFGF, SHH, NT3 and BDNF treatment of hiPSC-derived ONPs	ONPs give rise to sensory neuron-like cells Comparable results for LV-and mRNA-reprogrammed hiPSCs	[42]
Neurospheres + Attachment on human fibroblast feeders SFM + EGF + bFGF and posterior GF withdrawal	Bipolar neurons that express sensory neuron markers ISL1, BRN3A, NEUROD1, GATA3, VGLUT1 Functionally active hiPSC-derived neurons establish contact with IHCs and OHCs in cochlear explants. Synapsin 1 expression. Number of synapses dependent on differentiation stage of the neurons Compared to hESCs, higher variability in marker expression and lower potential of hiPSC-derived neurons to establish contact with denervated HCs in vitro	[47,48]
**SC-like Cells**		
**Differentiation Method**	**Main Outcomes**	**Reference**
SFEBq aggregates + BMP and/or Insulin Culture of CX26+ vesicles on cochlear feeder cells	Numbers of CX26-expressing cells promoted by insulin treatment and reduced when the TGFβ pathway is inhibited The CX26-expressing cells grown on feeder cells are proliferative, form gap junctions at cell-cell borders and express markers of cochlear SCs	[37]
Monolayer OPs: transient BMP signalling + continued FGF activation OSC: NaHCO_3_	Highly efficient method to obtain OPs and OSCs that display a mature OSC phenotype, with Pendrin expression and anion exchange activity	[36]
**OTHERS**		
**Differentiation Method**	**Main Outcomes**	**Reference**
LV-and mRNA-reprogrammed hiPSC lines Monolayer cultures FGF3 + FGF10 treatment	Two types of OPs (PAX8, PAX2, SOX2, FOXG1): OEPs and ONPs. These can be maintained in a proliferative state. OEPs give rise to HC-like cells ONPs give rise to SGN-like cells Comparable results for LV- and mRNA-reprogrammed hiPSCs	[42]
Monolayer + Signalling modulation + Single-cell gene expression analysis Panel of 90 genes analysed Comparisons to native otic cells from mouse otocyst	Conditions established for the stepwise induction of NNE/PPE/early otic lineage. - Suppression of mesodermal and endodermal lineages by blocking TGFb and WNT signalling (day 1–6). Upregulation of NNE markers (64% of the cells) - Asynchronous differentiation - Identification of NNE genes: vs. mesendoderm: TFAP2A, GATA3, DLX3, EYA1 (Mesendoderm markers: GATA4, Brachyury, POU5F1, ISL1, MSX1) vs. neural ectoderm: MSX2, TFAP2A, GATA3 (Neural ectoderm: LHX2, RAX, PAX6) - RA (day 6–8) induces expression of posterior placode genes PAX2 and PAX8 in NNE cultures. Stronger induction if NNE cultures exposed to RA (day 6–8) + bFGF (day 6–18) + BMP inhibitor (day 6–12) + BMP4/WNT (day 12–18). Yet, tendency of NNE to differentiate towards anterior fates (PAX6+) Delayed upregulation of PPE markers in hiPSC-compared to hESC-derived cultures. More robust and coordinated induction of otic markers in the latter. Comparison to mouse otocyst: hESC-and hiPSC-derived cultures exhibit the closest resemblance to native mouse otocyst cells at day 12 Single-cell analyses to identify factors that lead to an optimization of the culture conditions to differentiate stem cells to otic cell types	[49]
Monolayer + FGF signalling ± TGFβ and WNT inhibition	Rapid method to generate otic/placodal progenitors. Simultaneous FGF activation and TGFβ and WNT inhibition leads to higher rates of PAX2-positive cells and OP marker expression, as compared to FGF-only activation	[45,46]
**ORGANOIDS**		
**Differentiation Method**	**Main Outcomes**	**Reference**
2D culture and 3D culture systems Modulation by GFs in medium Hypoxia/Normoxia	Highly efficient method to generate homogenous OP cultures: 2D monolayers and controlled medium conditions SGN-like cells in the surface of otic organoids SGN-like cells with morphology, protein expression patterns and electrophysiological characteristics similar to primary SGNs. Presence of HC-like cells in the otic organoids	[50]
Rotary cell culture system	Highly robust and efficient protocol Large numbers of vestibular HC-like cells that resemble human foetal vestibular HCs. Heterogeneous HC-like cell population Otoconia-like structures Neuronal-like cells present in the cultures	[51]
Modifications to the method by Koehler et al., 2018: MEF feeders, cell number, low-adhesion plates, concentration and application times of mercaptoethanol, Matrigel	Vestibular and cochlear HC-like phenotypes. Immature types Electrophysiologically active HC-like cells Bipolar sensory neurons Possible synaptic contacts sensory neuron-/HC-like cells	[52]
Sequential treatment of aggregates BMP/TGFβ inh: NNE FGF/BMP Inh: Posterior placode, OEPD Self-guided differentiation in minimal medium + Matrigel (+Wnt)	Vestibular HC-like cells. Electrophysiologically active Unipolar and bipolar neurons Possible synaptic contacts sensory neuron-/HC-like cells Keratinocytes, putative Schwann cells, mesenchymal cells, chondrocytes present in the cultures	[31]

hiPSCs have been differentiated towards HC, SGN, SC and OP lineages. Inner ear organoids have been recently generated that harbour a variety of cell types and may thus be used to study different cell populations. The table briefly presents some data on the experimental set-up and the main findings in each of the referenced studies. hiPSCs: human induced pluripotent stem cells; HC: hair cell; SGN: spiral ganglion neuron; SC: supporting cell; OP: otic progenitor; LV: lentivirus; OEPs: otic epithelial progenitors; EBs: embryoid bodies; GF: growth factor; RA: retinoic acid; ONPs: otic neural progenitor; IHCs: inner hair cells; OHCs: outer hair cells; hESCs: human embryonic stem cells; SFEBq: serum-free floating culture of embryoid body-like aggregates with quick reaggregation; OSC: outer sulcus cell; NNE: non-neural ectoderm; PPE: preplacodal ectoderm; Inh: inhibitor; OEPD: otic-epibranchial placode domain.

**Table 2 cells-11-03331-t002:** List of hiPSC lines that have been generated from hearing-impaired patients.

Gene	Disorder	Donor Cell Type	Otic Cell Type	Mutated Phenotype	Corrected Phenotype	Ref.
** *MYO7A* ** **Compound heterozygous C.1184G > A and** **C.4118C > T**	Deafness	Urinary cells	OEPs ONPs HC-like cells	-Disarrayed, curved stereocilia with no links between them -MET channel dysfunction -Abnormal electrophysiological activity	-Organized and rigid stereocilia-like protrusions with links between them -Restored electrophysiological functions	[59]
** *MYO* ** **15A** **Compound heterozygous C.4642G > A and C.8374G > A**	Profound HL	Dermal fibroblasts	OP-like cells HC-like cells	-F-actin disorganization, shorter stereocilia and dysfunction of HC-like cells. Syncytia formation and some cell death during differentiation	-Rescue of F-actin organization, normal stereocilia length -Functional HC-like cells -No syncytia formation	[60]
** *TMC* ** **1** **Heterozygous dominant C.1253T > A**	Dominant non-syndromic HL	Urine cells				[90,95]
** *SMPX* ** **C.133–1G > A**	X-linked Deafness 4 (DFNX4)	PBMCs				[92]
** *USH* ** **2** **USH2A mutation C.8559–2A > G**	Usher syndrome type 2	PBMCs				[100]
** *USH* ** **2** **USH2A Compound heterozygous C.2299DELG and C.1256G > T**	Usher syndrome type 2	PBMCs				[94]
** *USH* ** **2** **USH2A** **Homozygous C.2299DELG** **Compound heterozygous C.2299DELG and C.2276G > T**	Usher syndrome type 2	PBMCs				[101,102]
** *PAX* ** **3** **Autosomal dominant heterozygous splice site mutation C.452–2A > G**	Waardenburg syndrome	Immortalized B lymphocytes				[104]
** *PAX* ** **3** **Autosomal dominant heterozygous frameshift mutation C.123DEL**	Waardenburg syndrome	PBMCs				[105]
** *SOX* ** **10** **Autosomal dominant heterozygous mutation C.336G > A**	Waardenburg syndrome	Dermal fibroblasts	NCCs			[69]
** *SOX* ** **10** **SINE-VNTR-ALU retrotransposon insertion into intron 2** **G.37982884_37982885 INS (3002)**	Waardenburg syndrome	PBMCs				[123]
** *ATP* ** **6V1B2** **Dominant heterozygous mutation C.1516C > T**	Dominant deafness-onychodystrophy syndrome	PBMCs				[124,125]
** *CISD* ** **2** **C.103 + 1G > A**	Wolfram syndrome	Fibroblasts				[126]
** *MAP* ** **1B** **C.4198A > G**	Profound non-syndromic HL	PBMCs	Otic sensory neuron-like cells	-Altered dynamics of microtubules and axonal elongation -Deficient electrophysiological activity		[62]
** *AIFM* ** **1** **C.1265G > A**	X-linked late-onset auditory neuropathy	PBMCs				[106]
** *GJB* ** **2** **Homozygous C.235DELC**	Non-syndromic HL	PBMCs	Connexin 26 gap junction-forming cells with characteristics of SCs	-Shortened gap junction plaque lengths -Deficient gap junction intercellular communication		[37,108]
** *GJB* ** **2** **C.109G > A**	Non-syndromic HL	PBMCs				[89]
** *GJB* ** **2** **Homozygous P.G45E/Y136X**	Non-syndromic HL	PBMCs				[68]
** *GJB* ** **2** **Compound heterozygous C.235DELC AND C.299–300DEL**	Non-syndromic HL	PBMCs				[109]
** *GJB* ** **2** **Homozygous mutation C.235DEL C**	Non-syndromic HL	PBMCs				[91]
** *GJB* ** **2** **Homozygous mutation C.109G > A**	Non-syndromic HL	PBMCs				[127]
** *GJB* ** **2** **Homozygous mutation C.109G > A**	Non-syndromic HL	PBMCs				[128]
** *SLC* ** **26A4** **Homozygous mutation C.1229C > T** **Homozygous mutation C.2168A > G** **Compound heterozygous C.439A > G and C.2168A > G**	Pendred Disease	Monocytes	OP-like cells OSC-like cells	-Anion exchange activity -Pendrin aggregates in the cytoplasm -Decreased clearance of mutant protein -High death rates following proteasome inhibition	-Anion exchange activity -Reduced numbers of cells containing pendrin aggregates -Cells less susceptible to cellular stress	[36,129]
** *SLC* ** **26A4** **Homozygous mutation C.919–2A > G**	Pendred Disease	PBMCs				[115]
** *SLC* ** **26A4** **Compound heterozygous C.919–2A > G AND C.1614 + 1G > A**	Pendred Disease	PBMCs				[114]
** *TMEM* ** **43** **Heterozygous dominant C.1114C > T**	Late-onset auditory neuropathy spectrum disorder	Lymphoblastoid cell line				[111]
** *ESRP* ** **1** **Compound heterozygous C.665_683 DEL AND C.775C > G**	Profound HL	Lymphoblastoid cell lines				[79]
** *P2RX* ** **2** **Heterozygous (and ho-mozygous) mutation C.178G > T**	Autosomal dominant progressive HL	Renal epithelial cells				[88]
** *GDF* ** **6** **Deletion in 3′ end of GDF6**	Non-syndromic HL associated with cochlear aplasia	Fibroblasts	Non-neuronal ectoderm and preplacodal ectoderm			[82]
** *MT-RNR* ** **1** **Mitochondrial A1555A > G**	Severe-to-profound non-syndromic HL	PBMCs				[93]
** *MT-RNR* ** **1/TRMU** **Mitochondrial A1555A > G** **±** **C.28G > T**	Phenotypic variability	Lymphoblastoid cell lines	Otic epithelial progenitors HC-like cell	-Deficient differentiation to otic epithelial progenitors -HC-like cells exhibit morphological and electrophysiological deficits		[119]
**MTDNA** A8344G	Myoclonus epilepsy associated with ragged-red fibres (syndromic HL)	Fibroblasts	HC-like cells	-Increased intracellular ROS and impaired ROS scavenging capacities of hiPSCs and hiPSC-derived HC-like cells -Defects in stereociliary bundles		[43]

The donor cell type is indicated. In those cases where the hiPSCs have been differentiated towards an otic lineage, this is indicated, as well as the observations that were made on the cultures carrying the mutation(s). In some cases, the hiPSCs were differentiated to more than one otic cell type. In this case, the cell types that exhibited an altered phenotype are indicated in bold. In some cases, an isogenic hiPSC line was created after correcting the original mutation. The mutations that were corrected to generate the isogenic lines are indicated in red, and the phenotype of the cell progenies of the genetically corrected hiPSC lines is indicated. OP: otic progenitor; OEPs: otic epithelial progenitors; ONPs: otic neural progenitors; HC: hair cell; HL: hearing loss; MET: mechanotransduction; PBMCs: peripheral blood mononuclear cells; NCCs: neural crest cells; OSC: outer sulcus cell; hiPSC: human induced pluripotent stem cells; SCs: supporting cells; ROS: reactive oxygen species.

## Data Availability

Not applicable.

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
