# Peer review of "Induced Pluripotent Stem Cells, a Stepping Stone to In Vitro Human Models of Hearing Loss"

_cells, 2022, doi:10.3390/cells11203331_

Round 1

Reviewer 1 Report

Duran Alonso reviewed the recent advances in the iPS-based hearing loss research.  Sufficient numbers of previous reports are covered, and the overall quality of the manuscript is quite high.  I can make some comments; if possible, please address them adequately.

Minor points:

1. In the introduction sessions, the authors mentioned zebrafish as a model animal as well as rodents model (L50). Zebrafish is also a good model animal; however, in this manuscript, they reviewed iPS cells as human models. Recently, primate model animals, such as the common marmoset, are come to be used as model animals in this field, including in developmental biology and genetic hearing loss.  Therefore, the authors should be mentioned this primate model. Moreover, they should add several paragraphs comparing the conventional rodent model, recent primate model (as primate model in vivo), and human iPS-based model (as primate model in vitro) with several citations. I believe this discussion would be helpful for the readers and increase the value of this manuscript.

2. Several paragraphs are too long to follow. Please, rewrite them.

Reviewer 2 Report

It was great pleasure reading the comprehensive review of hearing loss and iPSC (of course human). 

I feel this manuscript has good enough information to be published.

I do have some minor comments regarding the genetic mutation iPSCs.

As far as I know, not all researches included in this paper have developed organoid or differentiated hair cell like cell from hiPSC.

For example hiPSCs from TMC1 gene variant patients have been published, not sure this is from different patients since two studies (citation 82, 87) shares the same authors. However, no differentiation outcome from these cell line is published yet. Publications introduced in the manuscript including Beethoven mouse is mouse animal model. 

Therefore, I think author should comment or add information about outcomes of inner ear organoid generation or 2D cell differentiation from these hiPSC from patients with genetic variants which can result in hearing loss. 
